# Degradable Plasma-Polymerized Poly(Ethylene Glycol)-Like Coating as a Matrix for Food-Packaging Applications

**DOI:** 10.3390/nano13202774

**Published:** 2023-10-16

**Authors:** Maryam Zabihzadeh Khajavi, Anton Nikiforov, Maryam Nilkar, Frank Devlieghere, Peter Ragaert, Nathalie De Geyter

**Affiliations:** 1Research Unit Food Microbiology and Food Preservation, Department of Food Technology, Safety and Health, Ghent University, Coupure Links 653, 9000 Ghent, Belgium; frank.devlieghere@ugent.be (F.D.); peter.ragaert@ugent.be (P.R.); 2Research Unit Plasma Technology, Department of Applied Physics, Ghent University, Sint-Pietersnieuwstraat 41, 9000 Ghent, Belgium; anton.nikiforov@ugent.be (A.N.); maryam.nilkar@ugent.be (M.N.); nathalie.degeyter@ugent.be (N.D.G.)

**Keywords:** plasma polymerization, atmospheric-pressure plasma, food packaging, biodegradable polymer, PEG-like coating

## Abstract

Currently, there is considerable interest in seeking an environmentally friendly technique that is neither thermally nor organic solvent-dependent for producing advanced polymer films for food-packaging applications. Among different approaches, plasma polymerization is a promising method that can deposit biodegradable coatings on top of polymer films. In this study, an atmospheric-pressure aerosol-assisted plasma deposition method was employed to develop a poly(ethylene glycol) (PEG)-like coating, which can act as a potential matrix for antimicrobial agents, by envisioning controlled-release food-packaging applications. Different plasma operating parameters, including the input power, monomer flow rate, and gap between the edge of the plasma head and substrate, were optimized to produce a PEG-like coating with a desirable water stability level and that can be biodegradable. The findings revealed that increased distance between the plasma head and substrate intensified gas-phase nucleation and diluted the active plasma species, which in turn led to the formation of a non-conformal rough coating. Conversely, at short plasma–substrate distances, smooth conformal coatings were obtained. Furthermore, at low input powers (<250 W), the chemical structure of the precursor was mostly preserved with a high retention of C-O functional groups due to limited monomer fragmentation. At the same time, these coatings exhibit low stability in water, which could be attributed to their low cross-linking degree. Increasing the power to 350 W resulted in the loss of the PEG-like chemical structure, which is due to the enhanced monomer fragmentation at high power. Nevertheless, owing to the enhanced cross-linking degree, these coatings were more stable in water. Finally, it could be concluded that a moderate input power (250–300 W) should be applied to obtain an acceptable tradeoff between the coating stability and PEG resemblance.

## 1. Introduction

Food packaging is a vital process to provide food protection, assure food safety and quality, prolong the shelf life, and reduce food loss and waste. In recent years, the food industry has faced an ever-growing demand for minimally processed fresh-like foods with a reasonable shelf-life stability. One of the most common problems with fresh ready-to-use foods is microbial decay during the post-processing handling step. The high demand for these kinds of food products has implied the necessity of packaging materials with advanced properties that can help in many aspects of food sustainability, including the prevention of bacterial colonization and biofilm formation [1,2]. 

Among the different materials used for food packaging, polyethylene (PE) film is often used because of its excellent chemical resistance, good water-vapor barrier properties, high impact strength, good processability, and high cost-performance ratio. However, despite these outstanding features, PE film, itself, does not possess any antimicrobial activity; and, consequently, numerous studies have been conducted to examine potent approaches for obtaining antimicrobial PE films [3,4]. Methods for preparing antibacterial PE packaging can be divided into two categories. The first one is the incorporation of antimicrobial agents into the PE matrix itself, while the other approach is the use of surface coatings on top of PE films. Unfortunately, several problems are associated with the first method, such as the low thermal stability of the antibacterial agents, which causes issues during the PE extrusion process; the incompatibility of the agents with the polymer; and, most importantly, the possibility of inhibiting PE recycling depending on the nature of the antibacterial agents [5]. Consequently, techniques for coating surfaces on top of PE are preferable: antibacterial agents embedded in a controlled-solubility coating matrix in contact with water, allowing the slow release of the antibacterial agents over time, is considered as the most desirable coating type. In this case, the coating matrix can be entirely dissolved by the end of the product’s shelf life, which guarantees the recyclability of PE afterward [5,6]. 

Among the different polymers suitable for preparing such coating matrices, poly(ethylene glycol) (PEG), which is composed of repeating ethylene glycol units [-(CH_2_CH_2_O)_n_], is very interesting as PEG is nontoxic, biodegradable, biocompatible, and approved by the Food and Drug Administration (FDA) [7]. The biodegradability of PEG is also beneficial for the waste reduction of food-packaging material [8,9]. In addition, PEG possesses excellent antifouling properties, can protect surfaces from microbial attacks, and shows a high thermal and chemical stability over extended periods. Consequently, PEG can be safely used to prepare a coating matrix on PE food-packaging films [10]. In addition, antibacterial agents can also be embedded in this coating, thereby releasing these agents and providing antibacterial properties to the food packaging [11]. Unfortunately, the development of a PEG-based coating on PE is exceedingly challenging since PEG forms a viscous liquid or wax at ambient temperature. Various techniques for PEG coating production, including grafting [12] and self-assembled monolayer (SAM) formation [13], have been successfully developed to address the occurring restrictions. In this context, plasma polymerization, defined as the formation of polymeric coatings through the polymerization of monomer molecules in the plasma and deposition on a substrate, is also a very interesting approach [14]. In addition to being green, plasma polymerization holds the prospect of tuning coating properties, including the surface morphology, surface energy, chemical composition, and surface cross-linking. Moreover, plasma-deposited films generally possess excellent bonding and adhesion to substrates, and plasma polymerization allows superior control over film thickness growth, even at the nanometer range [15,16,17,18]. Plasma polymerization can also be applied to deposit nanostructured coatings, which can be used in different technological applications [19]. The thin nanoscale coatings and nanostructured surfaces produced through plasma polymerization can offer advantages over conventional microscale deposition, such as improved chemical and mechanical properties, water repellency, hardness, and corrosion resistance [20].

Different plasma-polymerization processes have already been developed to fabricate PEG-like coatings by making use of a capacitively coupled plasma chemical vapor deposition (CCP-CVD) process or a low-pressure dielectric barrier discharge [21,22]. The findings of these studies indicate that plasma polymerization can be used to deposit PEG-like coatings on diverse substrates, thereby achieving considerable coating uniformity, high reproducibility, and high chemical stability of the coatings. Nevertheless, these plasma processes require low pressure, which in turn hinders their widespread application, especially when considering inline processing. Despite the current progress, there is, thus, still room for developing less-complex plasma-polymerization systems that operate at atmospheric pressure. Because of the elimination of vacuum equipment in atmospheric-pressure plasmas, these processes are expected to be cheaper and more ecologically benign compared to low-pressure plasma systems [23,24,25,26,27].

Therefore, the current research intends to focus on the use of an atmospheric-pressure plasma system to produce PEG-like coatings by envisioning food applications. The present study utilizes an atmospheric-pressure aerosol-assisted plasma deposition (AAPD) approach to engineer a PEG-like coating from a tri(ethylene glycol) divinyl ether monomer on an ultrahigh-molecular-weight PE substrate. The influences of different plasma operating parameters, including power, monomer flow rate, and distance to the substrate are examined to obtain a PEG-like coating that can potentially act as a matrix to incorporate antibacterial agents for food-packaging applications. The chemistry and morphology of the coatings are extensively investigated using a wide set of surface characterization tools. To fulfil the requirements of a suitable matrix for the controlled release of embedded antibacterial agents, the plasma-polymerized PEG coating should also have a precisely controlled solubility in water. The water penetration of the matrix can be controlled by accurately tuning the plasma-polymerization parameters as they are known to affect the coating cross-linking degree and coating wettability, which both directly influence the extent of water diffusion inside the coating. It must be noted that the complete dissolution of the coating by the end of the product’s shelf life will provide additional value as it gives opportunities for PE recycling as an extra end-of-life option [28,29,30]. As such, particular attention is also paid, in this study, to the hydrolysis of the coatings in an aqueous environment because of its high importance in the controlled release of antibacterial agents.

## 2. Materials and Methods

### 2.1. Materials

Nitrogen (N_2_) (purity: ≥99.999%) working gas was purchased from Air Liquide (Brussels, Belgium). Tri(ethylene glycol) divinyl ether (DVE-3) (purity: ≥98%) was obtained from Sigma-Aldrich (Brussels, Belgium) and used as precursor for the plasma-polymerization process. The chemical structure of DVE-3 is presented in Figure 1. The applied substrate in this study was 75 µm thick ultrahigh-molecular-weight polyethylene (UHMWPE) purchased from the Goodfellow (Huntingdon, England). Before the plasma polymerization, the UHMWPE films were cut into 10 × 10 cm^2^ squares and washed with 2-propanol ≥99.5% obtained from Carl Roth (Karlsruhe, Germany). Single-side polished, 100 ± 0.3 mm diameter, prime-grade silicon wafers were supplied by Siegert Wafer GmbH (Germany) and used as deposition substrates for optimizing the atomizer gas-flow rate and for acquiring cross-sectional scanning electron microscopy (SEM) images.

### 2.2. Plasma Deposition

The PlasmaSpot^®^ 500 (Molecular Plasma Group (MPG), Foetz, Luxembourg), which is a dielectric barrier discharge (DBD) plasma system, was used for depositing the coatings and is schematically represented in Figure 2. The electrode configuration consisted of two concentric tubular electrodes: the inner electrode was grounded, while the external electrode was connected to an alternating current (AC) power source and covered by a dielectric layer on its inner side. The powered electrode was fed with high voltage up to 15 kVp-p, at a frequency in the range 50–60 kHz.

In this system, nitrogen was used as the feeding plasma gas at a fixed flow rate of 80 standard liters per minute (slm) and was sent to the 5 mm gap between the inner electrode and dielectric layer, as shown in Figure 2. In addition, a carrier gas for precursor injection was also used and sent to the hollow core of the inner electrode. The injection system introduced the precursor DVE-3 to the plasma afterglow in the form of aerosol droplets with a typical size of 50–150 nm (depending on the viscosity of the liquid and the atomizer gas-flow rate). Aerosol-assisted plasma deposition is a recently developed approach having the advantage of control over the precursor dissociation and over the coating cross-linking degree [15]. This approach has already been successfully applied to deposit antibacterial layers, coatings on heat-sensitive materials, and even coatings containing complex drugs for biomedical applications [16,17,18]. As depicted in Figure 2, a venturi-based atomizer equipped with a recirculation reservoir and an inlet for dilution gas was implemented for generating precursor aerosols. Nitrogen, at a fixed flow rate of 5 slm, was used as the dilution gas, and different atomizer gas-flow rates (0.6, 0.7, 0.8, 0.9, and 1.0 slm) were applied to transfer the formed aerosols through the core of the inner electrode. In each case, the coating was deposited by moving the plasma head over the substrate surface using a moving stage at a scan speed of 34 mm/s and a track width of 5 mm. Overall, eight passes were used to deposit coatings with thicknesses above 50 nm that allowed us to use various analytical techniques for chemical and morphological coating characterization. The use of more than eight passes was undesirable owing to excess deposition, leading to the formation of non-conformal layers. 

In the first step, the atomizer gas-flow rate was optimized by coating silicon wafers and examining the stability of the obtained deposits in contact with water. For this purpose, a silicon wafer instead of UHMWPE was used as the substrate as the disappearance of the plasma-polymerized PEG-like coating in contact with water could be easily visually monitored based on color differences. Moreover, the changes in the elemental composition of the coatings after immersion in water were also monitored via X-ray photoelectron spectroscopy (XPS), and the detection of the Si peak originating from the silicon substrate indicated the disappearance of the coatings. Appendix A lists the elemental compositional changes in the coatings deposited at different atomizer gas-flow rates. The silicon wafer was placed 2 mm from the edge of the plasma head, and an input power of 175 W was used for this optimization step. Based on the obtained stability results, an atomizer gas-flow rate of 0.6 slm was found to be optimal owing to the preservation of the initial elemental coating composition after 72 h of contact with water. All the films were, therefore, deposited at a fixed atomizer gas-flow rate of 0.6 slm. 

Using the optimized atomizer gas-flow rate, films were subsequently deposited on UHMWPE, the substrate of interest. To gain a thorough insight into the influence of the distance between the tip of the electrode and the substrate on the coating characteristics, UHMWPE was first placed at varying distances from the electrode tip (2, 10, and 20 mm) while maintaining a constant input power of 175 W. Based on the obtained results, the optimal plasma–substrate distance capable of producing uniform coatings with an elemental composition similar to those of conventional PEG polymers was selected. Afterward, coatings were deposited at various input powers (175, 200, 250, 300, and 350 W) at the optimized distance. Subsequently, the effect of the input power on the physicochemical properties of the obtained coatings was also profoundly investigated in this study.

### 2.3. Coating Characterization

After the plasma polymerization at different plasma operational parameters, the DVE-3-based coatings were physically and chemically evaluated using different surface characterization tools. The coating surface morphology (atomic force microscopy (AFM) and scanning electron microscopy (SEM)), surface chemistry (XPS and Fourier-transform infrared spectroscopy (FTIR)), and surface wettability (water contact angle (WCA) analysis) were carefully examined. The experimental procedures used for each of these techniques will be described hereafter.

#### 2.3.1. AFM

To compare the surface morphology and surface roughness of the bare and plasma-coated UHMWPE substrates, an XE-70 atomic force microscope (Park Systems, Sliedrecht, The Netherlands) operating in the non-contact mode with a silicon cantilever (Nanosensors^TM^ PPP-NCHR) was utilized. The scan size for all the measurements was 30 × 30 μm^2^. For each sample, three randomly chosen areas were imaged and analyzed. XEI software 5.2.0 (Park system, Yongin, Republic of Korea) was used to calculate the root-mean-square roughness (*Rq*). *Rq* expresses the profile height deviation from the mean line according to the following equation [31]:(1)$$Rq=\sqrt{\frac{\sum_{i=1}^{n}{\left(Z_{i}-Z_{avg}\right)}^{2}}{N}},$$
where *Z_i_* is the height of each point, *Z_avg_* indicates the mean-height distance, and *N* is the number of surface height data. These surface roughness values were obtained from three randomly chosen areas on three samples under each condition and are presented as mean values ± standard deviations in this work.

#### 2.3.2. SEM

In addition to AFM, SEM was used to visualize the surface morphology of the uncoated and plasma-coated UHMWPE substrates. For this purpose, top-view SEM images were taken using a JSM-6010PLUS SEM device (JEOL, Tokyo, Japan) operating at an acceleration voltage of 7 kV and a magnification of 3000×. Prior to SEM imaging, the samples were gold-coated for 40 s using a JFC-130 auto fine sputter coater (JEOL, Tokyo, Japan) to avoid charge accumulation on the samples. In addition, the same SEM device was used to obtain cross-sectional SEM images as these can provide valuable information on the coating thickness. For this purpose, an acceleration voltage of 7 kV was also used but in combination with a higher magnification of 20,000×.

#### 2.3.3. XPS

In addition to the surface morphology and roughness, the surface chemical composition of the uncoated and plasma-coated UHMWPE substrates was examined using XPS analysis. A 5000 Versaprobe II spectrometer (Physical Electronics (PHI), Feldkirchen, Germany) equipped with a monochromatic Al Kα X-ray source (hν = 1486.6 eV) operating at a power of 24 W was employed. During the measurement process, the XPS chamber was maintained at a pressure below 10^−6^ Pa. A hemispherical analyzer tilted 45° to the normal of the coating’s surface was used to detect the photoelectrons emitted from the surface. XPS analysis was carried out on four randomly selected points on a single sample under each condition. Survey scans were recorded at a pass energy of 187.85 eV (eV step = 0.8 eV) to inspect the different surface elements. In addition, to define the specific types and relative amounts of the surface chemical bonds, high-resolution XPS spectra (C1s, N1s, and O1s) were recorded at a pass energy of 23.50 eV (eV step = 0.1 eV). The elements detected on the surface of the coatings were quantified using Multipak software (V 9.6) by applying a Shirley background while considering the relative sensitivity parameters assigned by the XPS device manufacturer. All the spectra were charge-corrected based on the C-O constituent of the C1s peak at 286.6 eV for the coatings possessing a PEG-like composition and based on the C-C constituent of the C1s peak at 285.0 eV for all the other coatings. The approach used to correct for the charge effect was chosen in accordance with previously published works [32,33]. The curves of the high-resolution spectra were also fitted using the same Multipak software. For the deconvolution of the peaks, Gaussian–Lorentzian curve shapes (80–100% Gaussian) with a full width at half maximum set below 1.5 eV were applied to each line shape.

#### 2.3.4. Static WCA Analysis

To analyze the surface wettability, static WCA assessments were performed at room temperature using a Krüss Easy Drop goniometer (Hoeilaart, Belgium). Drops of distilled water with a volume of 2 μL were deposited on the sample surface, and Laplace–Young curve fitting was used to determine the drop profile. WCA analysis was performed at at least 10 randomly selected locations of a single sample, and the obtained WCA values are represented as mean values ± standard deviations.

#### 2.3.5. Coating Stability

To examine the stability of the DVE-3-based coatings in contact with water, the obtained samples were immersed in distilled water and maintained at room temperature under static conditions. Samples were removed from the water at specific time intervals (0, 24, 48, and 72 h) and dried in a vacuum oven for further analysis. The changes in the coating surface morphology and surface roughness over time were evaluated using AFM, as described in Section 2.3.1. Furthermore, the coating surface chemistry was analyzed using FTIR, as this technique can provide valuable information on the coating stability. FTIR analysis was conducted using a Bruker Tensor 27 spectrometer (Kontich, Belgium) equipped with a single-reflection attenuated total reflectance (ATR) accessory (MIRacle, Pike technology) and a mercury cadmium telluride (MCT) detector. FTIR spectra were acquired in the spectral region 4000–700 cm^−1^ by averaging 64 scans (at a resolution of 4 cm^−1^) per measurement. OPUS 6 software was used for analyzing the spectra and compensating the atmospheric vapor to correct the absorption of water vapor and carbon dioxide present in the ambient environment. FTIR analysis was conducted at three randomly selected spots on three different samples per condition, and the averaged FTIR spectra are shown in this work.

## 3. Results and Discussion

It is well known that the chemical compositions and physical features of plasma-polymerized coatings are considerably influenced by the applied plasma operating parameters, such as the input power, precursor flow rate, and distance between the edge of the plasma head and substrate [34,35,36,37]. As previously mentioned, the precursor flow rate (or so-called “atomizer gas-flow rate” in this study) was initially optimized and fixed at 0.6 slm. However, the plasma–substrate distance and plasma input power were the variables in this study, and their influences on the characteristics of the deposited coatings were comprehensively examined.

### 3.1. Effect of Plasma–Substrate Distance on the Physicochemical Properties of DVE-3-Based Coatings

During the first step, the plasma polymerization of DVE-3 was conducted at various distances between the tip of the electrode (or the edge of the plasma head) and the substrate (2, 10, and 20 mm) at a constant input power of 175 W, and the impact of this distance on the coating properties was carefully examined. The reasons for choosing an input power of 175 W were to prevent excessive precursor fragmentation and to preserve the DVE-3 chemical structure as much as possible during the coating deposition.

Figure 3 shows the AFM and SEM images (top view and cross-section) of the bare and plasma-coated UHMWPE at plasma–substrate distances of 2, 10, and 20 mm. Clearly, the uncoated UHMWPE substrate has a relatively rough surface. On the other hand, the plasma polymerization of the DVE-3 precursor at 2 mm clearly results in the deposition of a conformal coating. However, when the plasma–substrate distance is further increased, particularly to 20 mm, non-conformal and less-uniform coatings are deposited in a manner such that the coating morphology broadly resembles the substrate morphology.

The same trend could be observed in the top-view SEM images. These images indicate apparent differences in the morphology of the DVE-3 coatings obtained at different distances. By increasing the plasma–substrate distance, an “island-like” coating is deposited, and the UHMWPE is not homogeneously covered. These findings could be ascribed to the dilution of the chemically active plasma species by expanding the gap between the plasma source and substrate. Accordingly, the dilution of active plasma species can lead to a reduction in the plasma efficiency of the polymerization [38,39], which subsequently results in the formation of a non-conformal coating. Our observations agree well with those in previous research conducted by Korzec et al. [39]. These researchers visualized the plasma–substrate contact areas of different atmospheric-pressure plasma jets at different plasma–substrate distances and found that the resulting contact area was reduced for all the examined plasma configurations by increasing the plasma–substrate distance. Table 1 displays the calculated roughness values (*Rq*) obtained from the AFM images of the uncoated and coated substrates. Clearly, the pristine UHMWPE has a relatively rough structure (*Rq* = 231.7 ± 24.9 nm) with considerable variation over the sample surface, as evidenced by the elevated standard deviation. Conversely, the DVE-3-based coatings are significantly smoother, particularly the coatings deposited at 2 mm, for which the surface roughness is low (*Rq* = 61.3 ± 3.2 nm). However, the data in Table 1 clearly reveal that the coating surface roughness increases again with increasing plasma–substrate distance. The observed effect could be explained by the occurrence/absence of gas-phase nucleation [40]. When the plasma–substrate distance is short, polymerization mainly occurs on the surface of the substrate, which contributes to the formation of a smooth conformal coating with a nanoscale roughness. This obtained nanoscale roughness can be very beneficial for the ultimate application as a matrix for embedding antibacterial agents. One of the crucial parameters affecting the antibacterial efficiency is the contact surface area provided for the packaged food. In this regard, an increased surface area (resulting from the nanoscale roughness) will enhance the antibacterial activity due to the more pronounced interaction of the bacteria with the surface [41].

On the other hand, with increasing plasma–substrate distance, gas-phase nucleation becomes increasingly dominant, leading to the clustering of polymerized precursors in the gas phase rather than on the substrate [40]. Consequently, at elevated plasma–substrate distances, non-conformal coatings with higher surface roughness values will be mainly deposited. 

For a more precise analysis of the morphology and to obtain insight into the coating thickness, cross-sectional SEM images were also obtained. For this purpose, coatings were deposited on silicon wafers instead of UHMWPE. These cross-sectional images clearly indicate that an increase in the plasma–substrate distance results in the formation of non-uniform coatings. The DVE-3 coating deposited at 2 mm reveals a smooth morphology with a thickness of 2.21 ± 0.18 µm. On the other hand, by increasing the distance to 10 and 20 mm, the coatings become non-uniform with considerable variation in the thickness ranging from 0.3 µm to 2.6 µm, resulting in an average thickness of 1.21 ± 0.86 and 1.32 ± 0.88 µm, respectively. These findings further confirm the formation of non-conformal DVE-3-based coatings arising from the increased plasma–substrate distances. 

In addition to the surface morphology and roughness, a detailed assessment of the surface elemental composition was conducted using XPS analysis to discern the effect of the plasma–substrate distance on the surface chemical composition of the samples. The elemental compositions of the UHMWPE substrate and coatings prepared at different distances are presented in Table 2. As expected, the pristine UHMWPE mainly consists of carbon with some traces of oxygen originating from surface contamination during air exposure. The DVE-3-based coatings are mainly composed of carbon and oxygen, which are the two elements that construct the precursor. Intriguingly, a small amount of nitrogen was also detected in the coating deposited at 2 mm, which could originate from the nitrogen-feeding plasma gas. In a nitrogen-containing plasma, various nitrogen species can be present, including neutral atomic nitrogen (N), molecular nitrogen ions (N_2_^+^), and excited nitrogen molecules (N_2_(A^3^∑_u_^+^)). Among these species, neutral atomic nitrogen could be generated at an appreciable concentration [42]. In addition, neutral atomic nitrogen is considered to have a high chemical reactivity and can directly be incorporated into a growing coating surface during plasma polymerization [43]. However, N atoms have a finite lifetime of approximately 100 µs [44,45,46]. Therefore, at gas-flow rate of 80 L/min, as used in the PlasmaSpot system, N atoms cannot travel over long distances (the characteristic decay distance is ~1 mm), which explains why nitrogen was only detected on the surface of the coating deposited at a plasma–substrate distance of 2 mm [44]. By disregarding the small amount of incorporated nitrogen, the surface oxygen (≈36%) and carbon (≈62%) contents approximate those of the chemical composition of PEG when the shortest plasma–substrate distance is used. The surface oxygen content, however, sharply decreases with increasing plasma–substrate distance, while, in turn, the surface carbon content increases. These results indicate that the chemical structure of the plasma-polymerized DVE-3-based coatings strongly depends on the plasma–substrate distance. It should be noted that even at the increased plasma–substrate distance of 20 mm, a thick coating (approximately 1.00 µm) is still deposited on the substrate, and the detected carbon and oxygen contents still arise from the coating itself and not from the UHMWPE substrate, as confirmed by the previously provided thickness measurements.

To obtain a clearer perception of the specific nature of the carbon- and oxygen-containing functional groups on the coating surfaces, high-resolution C1s and O1s core-level spectra were comprehensively assessed and deconvoluted. Figure 4 depicts the high-resolution C1s peaks for the UHMWPE substrate and the high-resolution C1s and O1s peaks for the coatings deposited at different plasma–substrate distances. Because of its low intensity, the O1s peak of the UHMWPE substrate is not shown in Figure 4. Figure 4 reveals that the C1s spectrum of the UHMWPE substrate is dominated by a large peak at 285.0 eV, corresponding to C-C/C-H bonds, as expected considering the chemical structure of UHMWPE. According to previous studies, the C1s spectrum of the DVE-3-based coating deposited at 2 mm deconvolutes into five different peaks: a peak attributed to C-H/C-C bonds at 285.0 eV, a peak assigned to C-N bonds at 285.7 eV, a peak accredited to C-O bonds at 286.6 eV, a peak corresponding to C=O/N-C=O at 287.8 eV, and, finally, a peak ascribed to O-C=O groups at 289.1 eV. The O1s spectrum of this coating encompasses four peaks: a peak at 531.3 eV corresponding to N-C=O, a peak at 532.3 eV attributed to C=O groups, a peak at 532.8 eV assigned to C-O groups, and, finally, a peak at 533.7 eV accredited to O-C=O groups. On the other hand, for the DVE-3-based coatings deposited at longer plasma–substrate distances, the peaks corresponding to the nitrogen functional groups were not assigned owing to the absence of nitrogen in the surface elemental compositions. For these coatings, the high-resolution C1s spectrum predominantly has four peaks: C-H/C-C, C-O, C=O, and O-C=O, at these same binding energies. In addition, the O1s spectrum of the DVE-3-based coatings only encompasses three peaks, including C=O, C-O, and O-C=O [47,48]. When a DVE-3-based coating is deposited on the UHMWPE substrate at a plasma–substrate distance of 2 mm, the dominant peak in the C1s spectrum is no longer located at 285.0 eV (as is the case for UHMWPE) but at 286.6 eV because of the formation of new chemical groups (C-O bonds) originating from the monomer. In addition, a peak at 285.7 eV corresponding to C-N groups, a peak at 287.8 eV attributed to C=O/N-C=O bonds, and a peak at 289.1 eV accredited to O-C=O groups are detected. The obtained C1s spectrum, thus, agrees well with the XPS spectra of the plasma-deposited PEG-like coatings reported in other research papers [21,49], which indicates that the PEG polymer structure is retained when the plasma-polymerization process is conducted at a short plasma–substrate distance. However, when the plasma–substrate distance is increased, the relative contributions of the different C1s peaks noticeably changes such that the dominant peak in the C1s spectrum is at 285.0 eV, corresponding to C-C/C-H bonds.

When taking a closer look at the C1s curve-fitting results of the DVE-3-based coatings (Table 3), one can notice that the chemical structure of the coating deposited at 2 mm closely resembles that of the DVE-3 monomer, as evidenced by the high number of C-O functional groups in its structure. However, minor monomer fragmentation also occurs, as low amounts of other nitrogen- and oxygen-containing functional groups (C-N, C=O/N-C=O, and O-C=O) are on the surface of this sample. When the plasma–substrate distance is increased, the content of C-O functional groups drastically decreases from approximately 64 at.% to 11 at.%, suggesting that considerable monomer fragmentation occurs at increased distances. Simultaneously, the contents of the other oxygen-containing groups (C=O and O-C=O) also sharply decrease, which agrees with the previously shown elemental composition data. These results reveal that more enhanced monomer fragmentation occurs with increasing plasma–substrate distance, ultimately resulting in the deposition of a hydrocarbon-like coating containing only a minor content of oxidized groups at the longest plasma–substrate distance. By increasing the distance from the plasma source, there is, thus, sufficient time for precursor molecules to fragment, leading to a significant loss in the content of monomer C-O functional groups in the deposited coatings [38,50,51]. Short plasma–substrate distances are, thus, crucial to deposit a plasma-polymerized PEG-like coatings.

### 3.2. Effect of Input Power on the Physicochemical Properties of DVE-3-Based Coatings

In addition to the plasma–substrate distance, the input power is known to significantly affect coating properties when conducting plasma-polymerization experiments [15], and this impact will be studied in this section. Based on the influence of the plasma–substrate distance on the chemical and physical characteristics of the DVE-3-based coatings, a plasma–substrate distance of 2 mm was selected. This distance was chosen as it was, in this case, possible to deposit a conformal smooth coating while maintaining the C-O functional groups of the DVE-3 precursor. To comprehensively explore the influence of the input power on the coating properties, tests were performed at various input powers (175, 200, 250, 300, and 350 W) at a fixed plasma–substrate distance of 2 mm.

The cross-sectional SEM images of the UHMWPE substrate and the coatings deposited at various input powers are depicted in Figure 5. From these SEM images, considerable variations in the coating thickness were detected for different input powers. Based on the cross-sectional SEM images, it can be concluded that the coating thickness decreases with increasing input power. At low input powers (175 and 200 W), thick coatings are deposited at average thicknesses of 2.21 ± 0.18 and 2.02 ± 0.29 µm, respectively. However, further increases in the input power to 250 and 300 W result in more than a two-fold decrease in the coating thicknesses to 0.90 ± 0.06 and 0.74 ± 0.06 µm, respectively. Moreover, at the highest input power, 350 W, the coating thickness further reduced to 0.40 ± 0.13 µm. The observed thickness variation with increasing input power can be explained by the fact that the input power modifies the electron density and distribution of the electron energy, which, in turn, influences the chemical reactions occurring in the plasma environment, thereby changing the physicochemical properties of the deposited coatings [52,53]. It is well known that the input power per precursor molecule has a crucial impact on the extent of the precursor fragmentation as well as on the cross-linking degree of the deposited coating [34,54]. In conformity with the Yasuda model, when using a constant precursor flow rate, which is the case in this work, only a few monomer dissociation reactions occur at low input powers. Under these conditions, coatings are deposited at a high rate, and the obtained coatings primarily preserve the chemical structure of the precursor, resulting in a high retention of the precursor’s functional groups [55,56,57]. However, by increasing the input power per molecule, the plasma-polymerization mode can be switched to a monomer-deficient regime from a certain input power on. In this region, strong precursor fragmentation occurs, and competitive ablation and polymerization occur during the plasma-polymerization process [58,59]. As a result, coatings are deposited at a lower rate, resulting in thinner coatings that possess a higher cross-linking degree and a lower amount of the precursor’s functional groups. As revealed by the obtained thickness results, the DVE-based coatings were most likely obtained in the monomer-deficient regime at input powers ≥175 W when using a constant precursor gas-flow rate of 0.6 slm. 

The top-view SEM images and AFM images of plasma-polymerized coatings deposited at different input powers are shown in Figure 6. The top-view SEM images show that the plasma polymerization of DVE-3 at low input powers (175 and 200 W) provides a smooth and pinhole-free coating that homogeneously covers the entire substrate surface. However, considerable topographical changes can be observed with increasing input power: rough DVE-3-based coatings are deposited owing to the reduction in the coating thickness and the rough underlying substrate. These findings indicate that the application of high input powers results in thinner coatings that do not significantly change the morphology of the pristine UHMWPE. Therefore, under these conditions, the primary factor that determines the surface morphology of the obtained coatings is the substrate itself, which has a rough structure. This finding agrees with the AFM images, which also reveal that the surface roughness of the deposited coatings increases with increasing input power. 

As presented in Table 4, the surface roughness increases owing to the thickness reduction when the input power increases. However, the roughness of the coating obtained at the maximum input power of 350 W (*Rq* = 143.9 ± 2.5 nm) is still considerably lower than the roughness of the uncoated UHMWPE (*Rq* = 231.7 ± 24.9 nm), which confirms that a DVE-3-based coating is deposited on the surface, even at the highest used input power.

As in the previous section, XPS analysis was conducted to determine the influence of the input power on the chemical structure of the DVE-3-based coatings. The surface elemental compositions of the DVE-3 coatings deposited at different input powers are shown in Table 5. The coating prepared using an input power of 175 W primarily comprises carbon, oxygen, and a small amount of nitrogen. Carbon and oxygen are the two elements that are present in the precursor structure; however, this is not the case for nitrogen. As previously mentioned, the detected nitrogen most likely originates from the nitrogen-feeding plasma gas. The PlasmaSpot device operates in N_2_ gas, and neutral atomic nitrogen can be generated at high concentrations [43] and be directly incorporated into the deposited coating. The data listed in Table 5 also reveal that with increasing input power, the surface oxygen content decreases from approximately 36.3 at.% to 29.3 at.%, while the nitrogen content drastically increases from 1.9 at.% to almost 20.0 at.%. Simultaneously, the carbon content also decreases by more than 10 at.% when the input power is increased from 175 to 350 W.

The detected increase in the surface nitrogen content with increasing input power could be explained by reactions between nitrogen molecules and electrons, both of which are present in the nitrogen plasma [42]:N_2_ + ē → 2 N + ē(2)
N_2_ + ē → N_2_* + ē(3)

The second reaction generates excited nitrogen molecules, which can, in turn, react with molecular nitrogen, thereby leading to the formation of additional atomic nitrogen [42]:N_2_* + N_2_ → N + N + N_2_(4)(* denotes N_2_ (X^1^∑_g_^+^, V), N_2_ (A^3^∑_u_^+^), N_2_ (B^1 3^∑_g_^−^), and N_2_ (C^3^Π_u_)).

The dissociation reactions of nitrogen molecules to atomic nitrogen are intensified with increasing input power, which is expected to lead to an increased atomic nitrogen density in the gas phase. Subsequently, at higher input powers, more atomic nitrogen can be incorporated into the growing coating and form nitrogen-containing functional groups on the surface, as confirmed by the obtained XPS results. In addition to the changes in the nitrogen content, significant differences in the carbon and oxygen contents of the coatings are observed at different input powers. It is well known that with increasing input power, a higher concentration of reactive plasma species and more highly energetic species are present in the plasma, thereby resulting in more enhanced monomer fragmentation. As such, it is reasonable to assume that increasing the input power will intensify the DVE-3 fragmentation in the plasma, thereby leading to significant changes in the elemental composition of the coatings. 

The high-resolution C1s and O1s spectra were also deconvoluted to quantify the relative concentrations of the different surface functional groups in the coatings. The curve-fitted peaks obtained for different input powers are depicted in Figure 7, and similar peaks as those used for deconvoluting the C1s and O1s spectra of the DVE-3-based coatings deposited at a plasma–substrate distance of 2 mm (input power of 175 W) are used for deconvoluting these spectra. The relative concentrations of the carbon- and oxygen-containing functional groups, as obtained from the deconvoluted C1s and O1s spectra, are summarized in Table 6. 

Clearly, by increasing the input power, the relative concentrations of the carbon-containing functional groups substantially change. First, the relative C-O concentration gradually decreases with increasing input power, with only a few C-O groups remaining in the coating deposited at the highest input power (23.2%). This significant loss of C-O functional groups that are present in the DVE-3 chemical structure confirms that more pronounced monomer fragmentation occurs with increasing input power. On the other hand, the relative concentrations of other oxygen-containing functional groups, such as C=O and O-C=O groups, increase with increasing input power. Most likely, the C-O functional groups in the DVE-3 monomer are mainly destroyed at higher input powers, resulting in small monomer fragments, which can, in turn, recombine to deposit a coating containing higher relative contents of C=O and O-C=O groups [55,56,57]. These results agree with those previously published in the literature, as other studies have also reported the incorporation of oxygen-containing groups, such as carboxyl (O-C=O) and carbonyl (C=O) groups, on the surfaces of growing coatings during atmospheric-pressure plasma polymerization [60]. However, it is still important to highlight that despite the detected increase in the relative concentrations of the C=O and O-C=O groups, the surface oxygen content considerably decreases with increasing input power, meaning that the loss of C-O functional groups is much more pronounced compared to the increase in the contents of carbonyl and carboxyl functional groups. When looking at the relative concentrations of the oxygen-containing functional groups, similar conclusions can be drawn as those for the curve-fitted results of the C1s peaks: with increasing input power, the relative content of the C-O groups sharply decreases, while the relative contents of the C=O and O=C-O groups increase. 

The data listed in Table 6 also reveal that there is a large increase in the relative concentration of the C-N functional groups with increasing input power, which agrees with the observed increase in the nitrogen content, as presented in Table 5. As previously mentioned, more atomic nitrogen is incorporated into the coating with increasing input power, and this incorporation is mostly in the form of C-N bonds. However, amide groups are also present in the coatings, as revealed by the O1s curve-fitting results, and their concentrations also increase with increasing input power. Therefore, by increasing the input power, more monomer fragmentation occurs, resulting in a significant loss of C-O functional groups. As a result, at high input powers, completely different coatings containing a mixture of C-O, C=O, O-C=O, N-C=O, and C-N functional groups are deposited, and the chemical structures of the coatings bear no resemblance to that of PEG.

In addition to examining the influence of the input power on the surface morphology and surface chemical composition of the coatings, its influence on the surface wettability is examined in the following section. The WCA results obtained for various input powers are shown in Figure 8.

As expected, the uncoated UHMWPE substrate is hydrophobic, with a WCA value of 116 ± 1.7°, while the wettability of the DVE-3 plasma-polymerized coatings varies from hydrophilic (WCA value of 47 ± 0.8°) to highly hydrophilic (WCA value of 7.9 ± 1.2°) depending on the applied plasma input power: the higher the used input power, the higher the surface wettability and the lower the obtained WCA value. The observed decrease in WCA values with increasing input power can be attributed to large changes both in the surface chemical composition and surface roughness, as already evidenced in the previous sections focusing on the XPS analysis and SEM/AFM measurements. The XPS results clearly revealed the enhanced incorporation of nitrogen-containing functional groups (C-N and O-C=N) at higher input powers combined with enhanced contents of C=O and O-C=O functional groups at the expense of C-O groups. It is already well known that the presence of these nitrogen- and oxygen-containing functional groups on a coating’s surface can strongly enhance the coating’s hydrophilicity [61,62], which is also the case in this study. Another possible explanation for the observed WCA dependence on the input power could be related to the observed changes in the surface morphology. In the case of hydrophilic surfaces, it is well known that the apparent contact angle decreases when the solid’s surface roughness increases [63,64]. The obtained nanoscale roughness enhances the solid–liquid interfacial area, leading to improved wettability properties and a lower WCA value. The AFM results in Table 4 clearly show that the surface roughness increases with increasing input power, and this change can, in turn, contribute to the lower WCA values observed at higher input powers. Therefore, it is possible to deposit coatings possessing completely different chemical and physical properties, starting from DVE-3, by simply varying the input power used during the plasma polymerization. 

### 3.3. Water Stability

As it is very important for practical applications that developed coatings possess sufficient stability in water, this aspect was also closely examined in this work. In an initial stability study, the DVE-3-based coatings prepared using different input powers (at a fixed distance of 2 mm) were found to show very different stability behaviors after being immersed in water. In the case of the coatings deposited at low input powers (175 and 200 W), the coatings almost completely vanished within one hour of contact with water. On the other hand, when raising the input power, the stability of the coatings was improved. These observations can be explained by the enhanced monomer fragmentation and the accompanying higher cross-linking degree of the coatings when applying higher input powers. At lower input powers, the monomer fragmentation is not very pronounced, as was also evidenced by the chemical resemblance between the obtained coatings and the DVE-3 monomer molecules (see XPS section). In turn, the coatings deposited at low input powers are not highly cross-linked, resulting in poor coating stability in water. Based on the preliminary stability findings, an input power of 300 W was selected to investigate in more detail the performance of this DVE-3-based coating in contact with water. 

The FT-IR spectra of the DVE-3-based coating deposited at 300 W before and after contact with water for different time intervals are shown in Figure 9 for the wavenumber range 1900–950 cm^−1^. As shown, all the spectra are dominated by the same three major peaks (excluding the peak attributed to the substrate at 1470 cm^−1^). The spectra for all the samples contain a very broad band in the region 1000–1200 cm^−1^, which can be attributed to C-O stretching vibrations. The absorption band at 1633 cm^−1^ can be assigned to the amide I band, resulting from the stretching vibration of carbonyl groups (-C=O) in amide I structures [65]; while the peak at 1724 cm^−1^ can be attributed to carbonyl groups (C=O) in the structure of the PEG-like coatings. These results agree with the XPS analysis as similar chemical groups have also been identified using this technique [66,67]. As shown in Figure 9, the intensity of these three peaks decreases with increasing immersion time, while the intensity of the substrate peak increases. This observation can be explained by the gradual dissolution of the deposited coating during water immersion, which leads to a reduction in coating thickness with increasing immersion time. However, it is important to point out that even after the coating was in contact with water for 72 h, the peaks attributed to the DVE-3 coating are still visible in the measured FTIR spectra, confirming that a thin layer of the coating remains on the UHMWPE substrate after 3 days of water immersion. 

As the surface morphology and roughness can also provide some information about the coating stability, AFM measurements were conducted on the coatings deposited at 300 W for varying water immersion times, and the resulting AFM images are presented in Figure 8. As visualized in Figure 8, the surface becomes smoother after 24 h of contact with water, which is also evidenced by the assessed surface roughness, which decreases from 112.6 ± 2.4 nm to 79.7 ± 3.6 nm. This suggests that upon the initial water immersion, the coating surface becomes smoother with fewer cavities due to the removal of the top-layer of the coating. It is well known that this top layer is mechanically weak and can be easily abraded [67]. When the immersion time is further increased to 48 and 72 h, the coating surface becomes rougher again and gradually increasingly resembles the morphology of the uncoated UHMWPE substrate. Indeed, the *Rq* value raises to 113.7 ± 4.0 nm after 48 h of water exposure and, finally, even reaches 152.1 ± 6.8 nm after 72 h of exposure to water. However, it is worth mentioning that the roughness of the coating after 72 h of contact with water is still considerably less than the roughness of the UHMWPE substrate, suggesting that a part of the DVE-3-based coating remains on the substrate even after 72 h of water immersion. It can, thus, be concluded that the AFM measurements agree with the FTIR results, which also indicated the partial dissolution of the deposited DVE-3-based coating upon water immersion. These findings reveal that the DVE-3-based coatings developed in this study may act as a promising matrix for embedding antibacterial agents since it can allow the controlled release of the agents over time, owning to its gradual solubility in water.

## 4. Conclusions

The present study introduces a novel aerosol-assisted atmospheric-pressure plasma deposition technique to deposit plasma-polymerized DVE-3-based coatings on top of PE with potential applications in food packaging. The impacts of the most important plasma-polymerization parameters (plasma–substrate distance and monomer flow rate) on the physicochemical properties of the coatings was examined in detail. 

First, the DVE-3-based coatings were deposited on UMWPE at various plasma–substrate distances and a constant input power of 175 W. XPS, SEM, and AFM results revealed that a very short plasma–substrate distance of 2 mm was critical to achieve a smooth, conformal DVE-3-based coating with a chemical composition similar to those of typical PEG polymers.In the second step, the deposition process was conducted using different input powers while maintaining the plasma–substrate distance at 2 mm. The surface analysis results indicated a clear correlation between the applied input power per molecule and the extent of precursor fragmentation on the one hand and the coating stability in water on the other hand.When applying low input powers, hydrophilic DVE-3-based coatings were deposited, which chemically resembled PEG. Unfortunately, these coatings exhibited a very low water stability, which hindered their application as food-contact materials.Conversely, at high input powers, stable coatings in contact with water were obtained and were found to be highly hydrophilic owing to the extended incorporation of a wide range of oxygen- and nitrogen-functional groups on the surface of the deposited coatings. Although these DVE-3-based coatings do not chemically resemble PEG anymore owing to excessive monomer fragmentation, they do possess excellent potential to act as a matrix for embedding antibacterial agents. Such coatings can gradually release these agents over time in contact with food owing to their solubility in water.Overall, the aerosol-assisted non-thermal plasma technique studied in this work can, thus, be regarded as a promising green approach to develop degradable DVE-3-based coatings with immense control over their chemistry, morphology, and stability.

## 5. Outlook

In summary, the plasma-polymerization process can give us immense control over coating chemistry, morphology, and stability. Hydrophilic coatings chemically resembling PEG can be deposited when applying low input powers; unfortunately, these coatings were found to be highly unstable in water, making it impossible to use these coatings for food-packaging applications. On the other hand, stable, highly hydrophilic coatings could be obtained when applying high input powers. Unfortunately, these coatings do not chemically resemble PEG anymore owing to excessive monomer fragmentation. Nevertheless, these coatings could still be used in food-packaging applications when antibacterial agents are embedded in these coatings. In future work, the degradability of the stable coatings will be further examined, and their potential to act as a matrix for antibacterial agents will be explored in detail. 

## Figures and Tables

**Figure 1 nanomaterials-13-02774-f001:**
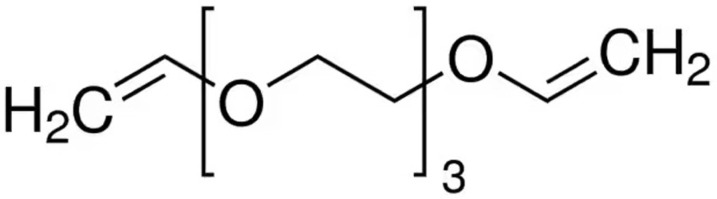
The chemical structure of tri(ethylene glycol) divinyl ether (DVE-3).

**Figure 2 nanomaterials-13-02774-f002:**
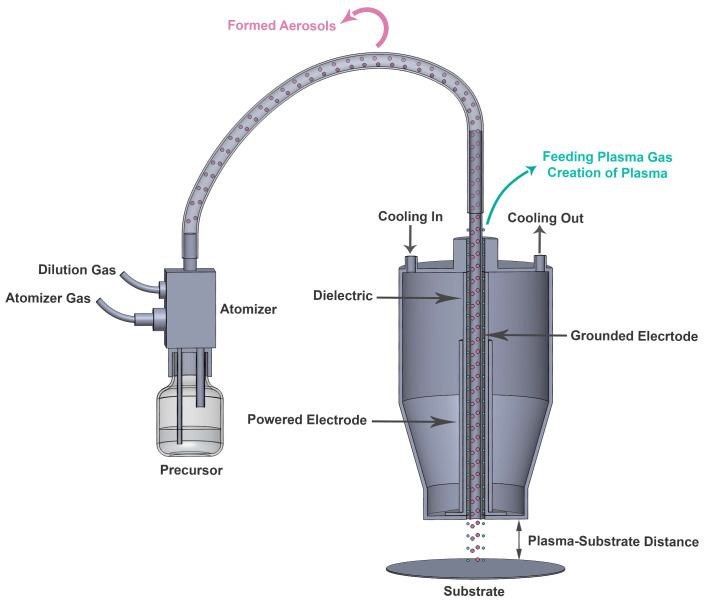
Schematic diagram of the atmospheric-pressure PlasmaSpot from MPG.

**Figure 3 nanomaterials-13-02774-f003:**
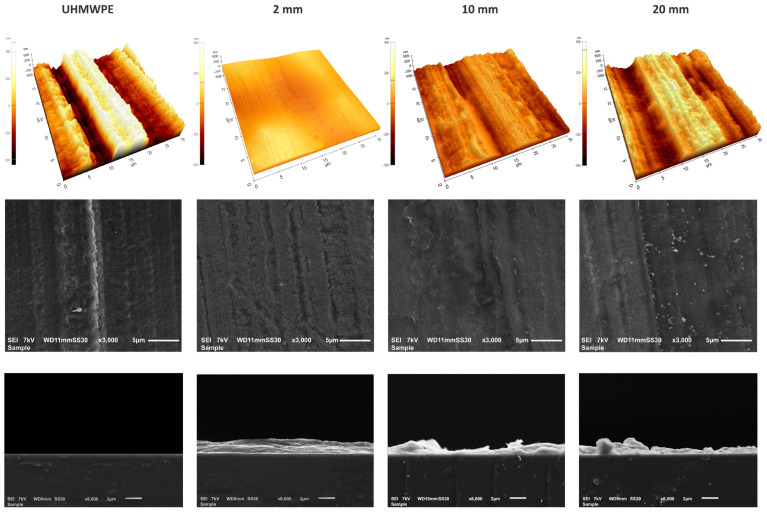
AFM and SEM images (top view and cross-section) from the surfaces of the UHMWPE substrate and the plasma-polymerized coatings for different plasma-to-substrate distances.

**Figure 4 nanomaterials-13-02774-f004:**
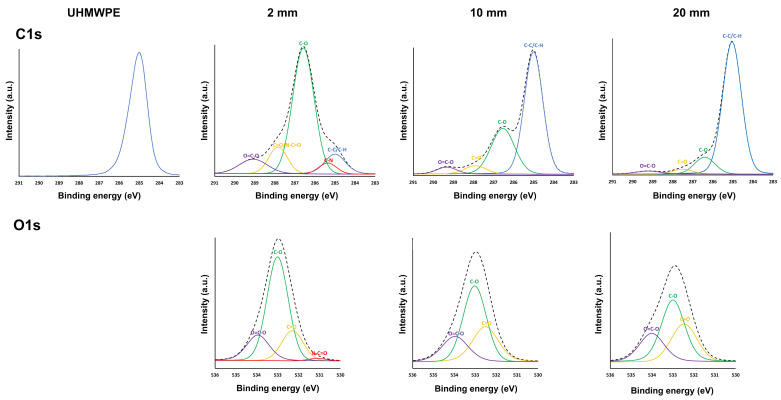
XPS high-resolution C1s and O1s peak fittings for plasma-polymerized coatings deposited at different plasma–substrate distances and the high-resolution C1s spectrum of UHMWPE.

**Figure 5 nanomaterials-13-02774-f005:**
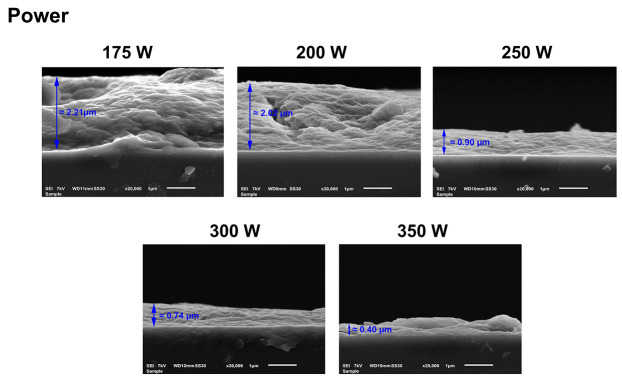
Cross-sectional SEM images of plasma-polymerized coatings deposited at different input powers.

**Figure 6 nanomaterials-13-02774-f006:**
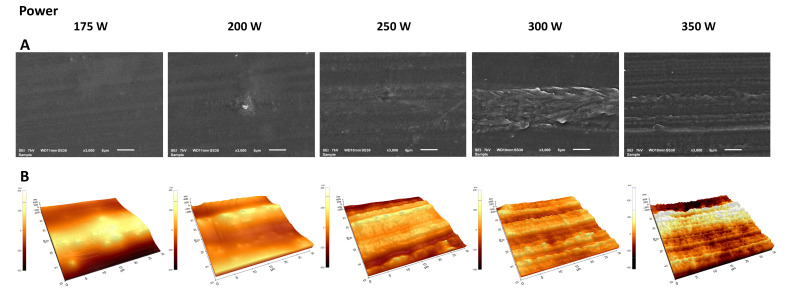
Top-view SEM images (**A**) and AFM images (**B**) of plasma-polymerized coatings deposited at different input powers.

**Figure 7 nanomaterials-13-02774-f007:**
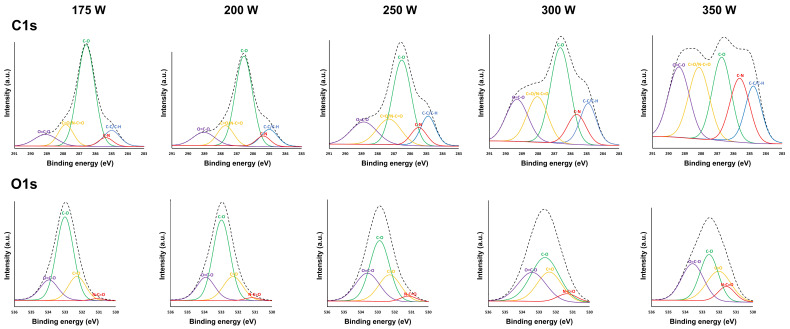
XPS high-resolution C1s and O1s peak fittings for plasma-polymerized coatings deposited at different input powers.

**Figure 8 nanomaterials-13-02774-f008:**
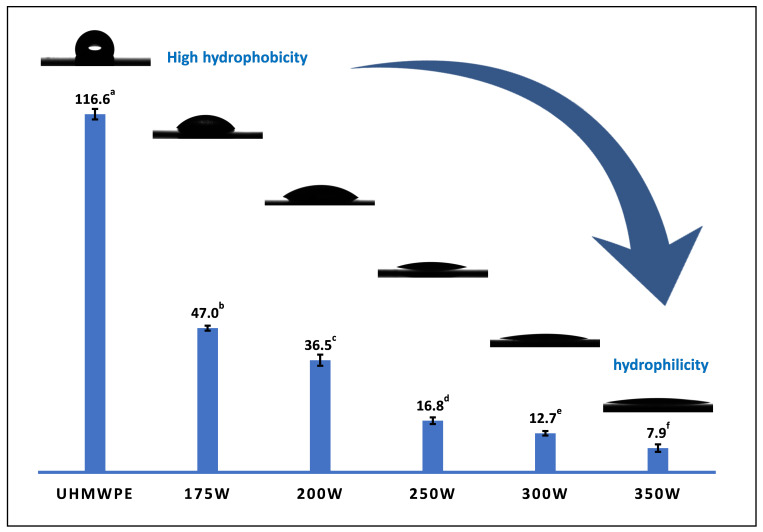
WCA evolution on UHMWPE polymer and plasma-polymerized coatings deposited at different input powers. ^a–f^ Values with different lowercase superscript letters are significantly different (*p* < 0.05) from each other.

**Figure 9 nanomaterials-13-02774-f009:**
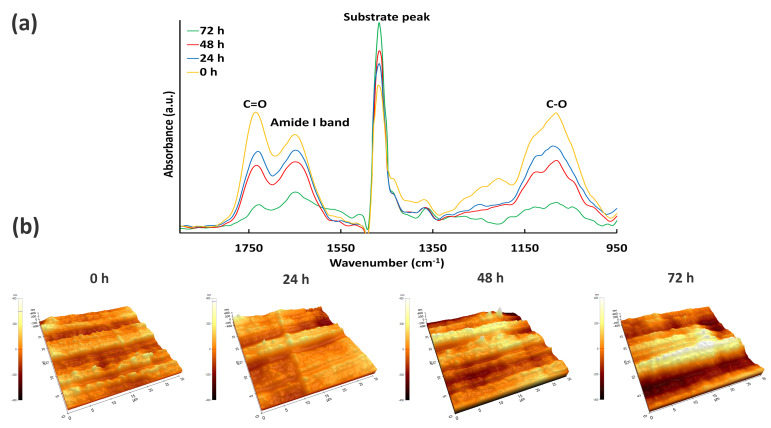
FT-IR spectra (**a**) and AFM images (**b**) of the plasma-polymerized coatings obtained at an input power of 300 W and in contact with water for different times.

**Table 1 nanomaterials-13-02774-t001:** Surface roughness values (*Rq*) for the UHMWPE substrate and plasma-polymerized DVE-3 coatings deposited at different plasma-to-substrate distances.

Sample	*Rq* (nm)
UHMWPE	231.7 ^a^ ± 24.9
2 mm	61.3 ^d^ ± 3.2
10 mm	97.0 ^c^ ± 10.7
20 mm	169.4 ^b^ ± 3.1

^a–d^ Values with different lowercase superscript letters are significantly different (*p* < 0.05) from each other.

**Table 2 nanomaterials-13-02774-t002:** Surface elemental compositions obtained using XPS for the UHMWPE substrate and plasma-polymerized DVE-3 coatings deposited at different plasma-to-substrate distances.

Sample	C (at.%)	O (at.%)	N (at.%)
UHMWPE	99.3 ^a^ ± 0.1	0.7 ^d^ ± 0.1	-
2 mm	61.9 ^d^ ± 0.8	36.3 ^a^ ± 0.5	1.9 ± 0.3
10 mm	80.4 ^c^ ± 1.6	19.6 ^b^ ± 1.6	-
20 mm	89.2 ^b^ ± 1.1	10.8 ^c^ ± 1.1	-

^a–d^ Columnar values with different lowercase superscript letters are significantly different (*p* < 0.05) from each other.

**Table 3 nanomaterials-13-02774-t003:** Deconvoluted C1s and O1s peaks of the high-resolution XPS spectra for plasma-polymerized coatings deposited at different plasma-to-substrate distances.

Sample	Peak	Assignment	Binding Energy (eV)	FWHM	Fitted Peak Area (%)
2 mm	C1s	C-C/C-H	285.0	1.22	10.7 ± 0.4
		C-N	285.7	1.03	4.6 ± 1.9
		C-O	286.6	1.26	63.7 ± 0.3
		C=O/N-C=O	287.8	1.09	11.7 ± 0.2
		O=C-O	289.1	1.63	9.4 ± 0.2
	O1s	N-C=O	531.2	1.00	1.6 ± 0.1
		C=O	532.3	1.22	18.5 ± 0.1
		C-O	532.8	1.19	62.1 ± 0.3
		O=C-O	533.7	1.28	17.9 ± 0.1
10 mm	C1s	C-C/C-H	285.0	1.09	62.5 ± 0.3
		C-O	286.5	1.38	29.2 ± 0.1
		C=O	287.9	1.27	4.7 ± 0.2
		O=C-O	289.2	1.17	3.6 ± 0.1
	O1s	C=O	532.4	1.45	28.6 ± 0.1
		C-O	532.9	1.29	50.2 ± 0.4
		O=C-O	533.9	1.38	21.1 ± 0.2
20 mm	C1s	C-C/C-H	285.0	1.10	84.3 ± 0.2
		C-O	286.6	1.20	11.3 ± 0.2
		C=O	287.7	1.16	2.8 ± 0.1
		O=C-O	289.2	1.09	1.6 ± 0.1
	O1s	C=O	532.5	1.45	29.4 ± 0.4
		C-O	533.0	1.31	47.1 ± 0.6
		O=C-O	534.0	1.41	23.5 ± 0.2

**Table 4 nanomaterials-13-02774-t004:** Roughness values (*Rq*) of the plasma-polymerized DVE-3 coatings deposited at different input powers.

Sample	*Rq* (nm)
175 W	61.3 ^e^ ± 3.2
200 W	72.6 ^d^ ± 3
250 W	84.1 ^c^ ± 4.4
300 W	112.6 ^b^ ± 2.4
350 W	143.9 ^a^ ± 2.5

^a–e^ Values with different lowercase superscript letters are significantly different (*p* < 0.05) from each other.

**Table 5 nanomaterials-13-02774-t005:** Surface elemental compositions of the plasma-polymerized DVE-3 coatings deposited at different input powers.

Sample	C (at.%)	O (at.%)	N (at.%)
175 W	61.9 ^a^ ± 0.8	36.3 ^a^ ± 0.5	1.9 ^d^ ± 0.3
200 W	61.7 ^a^ ± 1.3	35.2 ^ab^ ± 1.1	3.1 ^d^ ± 0.3
250 W	61.1 ^a^ ± 1.2	32.9 ^c^ ± 0.7	5.9 ^c^ ± 0.8
300 W	55.9 ^b^ ± 0.8	34.0 ^bc^ ± 0.6	10.1 ^b^ ± 0.9
350 W	51.1 ^c^ ± 1.7	29.3 ^d^ ± 1.7	19.6 ^a^ ± 3

^a–d^ Columnar values with different lowercase superscript letters are significantly different (*p* < 0.05) from each other.

**Table 6 nanomaterials-13-02774-t006:** Deconvoluted C1s and O1s peaks of the high-resolution XPS spectra for plasma-polymerized coatings deposited at different input powers.

Sample	Peak	Assignment	Binding Energy (eV)	FWHM	Fitted Peak Area (%)
175 W	C1s	C-C/C-H	285.0	1.22	10.7 ± 0.4
		C-N	285.7	1.03	4.6 ± 1.9
		C-O	286.6	1.26	63.7 ± 0.3
		C=O/N-C=O	287.8	1.09	11.7 ± 0.2
		O=C-O	289.1	1.63	9.4 ± 0.2
	O1s	N-C=O	531.2	1.00	1.6 ± 0.1
		C=O	532.3	1.22	18.5 ± 0.1
		C-O	532.8	1.19	62.1 ± 0.3
		O=C-O	533.7	1.28	17.9 ± 0.1
200 W	C1s	C-C/C-H	285.0	1.22	10.7 ± 0.3
		C-N	285.7	1.03	6.0 ± 1.5
		C-O	286.6	1.24	58.5 ± 0.2
		C=O/N-C=O	287.7	1.18	12.4 ± 0.5
		O=C-O	289.0	1.58	10.9 ± 0.3
	O1s	N-C=O	531.2	1.05	2.1 ± 0.1
		C=O	532.3	1.36	20.0 ± 0.4
		C-O	533.0	1.19	58.2 ± 0.7
		O=C-O	533.9	1.28	19.7 ± 0.3
250 W	C1s	C-C/C-H	284.9	1.09	13.7 ± 0.3
		C-N	285.6	1.16	8.5 ± 0.2
		C-O	286.5	1.35	44.5 ± 0.4
		C=O/N-C=O	287.3	1.74	17.2 ± 0.3
		O=C-O	288.9	1.89	16.2 ± 0.1
	O1s	N-C=O	531.2	1.14	3.9 ± 0.2
		C=O	532.3	1.53	23.2 ± 0.3
		C-O	532.9	1.36	46.5 ± 0.4
		O=C-O	533.6	1.68	26.4 ± 0.5
300 W	C1s	C-C/C-H	284.9	1.20	14.8 ± 0.2
		C-N	285.6	1.35	11.1 ± 0.3
		C-O	286.6	1.43	37.3 ± 0.7
		C=O/N-C=O	288.0	1.50	18.6 ± 0.2
		O=C-O	289.3	1.59	18.3 ± 0.3
	O1s	N-C=O	531.3	1.42	6.3 ± 0.2
		C=O	532.4	1.71	24.6 ± 0.3
		C-O	532.6	1.96	41.9 ± 0.3
		O=C-O	533.4	1.94	27.2 ± 0.6
350 W	C1s	C-C/C-H	284.8	1.20	15.1 ± 0.2
		C-N	285.7	1.42	18.4 ± 0.4
		C-O	286.7	1.37	23.2 ± 0.2
		C=O/N-C=O	288.1	1.50	21.7 ± 0.2
		O=C-O	289.4	1.52	21.6 ± 0.4
	O1s	N-C=O	531.6	1.19	10.3 ± 0.2
		C=O	532.0	1.71	27.7 ± 0.3
		C-O	532.6	1.28	31.8 ± 0.5
		O=C-O	533.6	1.54	30.2 ± 0.5

## Data Availability

All the data generated and analyzed during this study are included in this paper and the Appendix A. The data presented in this study are available on request from the corresponding author.

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
