# Peer review of "Degradable Plasma-Polymerized Poly(Ethylene Glycol)-Like Coating as a Matrix for Food-Packaging Applications"

_nanomaterials, 2023, doi:10.3390/nano13202774_

Round 1

Reviewer 1 Report

In this manuscript “Degradable plasma-polymerized poly(ethylene glycol)-like coatings as a matrix for food packaging applications”, the authors developed PEG-based coating using an atmospheric pressure aerosol-assisted plasma deposition method, which can act as a potential matrix for antimicrobial agents envisioning controlled-release food packaging applications. The authors optimized different plasma operating parameters including input power, monomer flow rate, and the gap between plasma head edge and the substrate to fabricate a PEG-like coating with a desirable water stability level.

Overall, the article looks novel and interesting. However, some shortcomings need to be addressed before possible publication in polymers.

Please find the attached annotated file to see my comments.

Based on my comments, the recommendation is Major Revision

Please my comments in the annotated PDF file

Reviewer 2 Report

The paper explores an eco-friendly method for producing advanced polymer films for food packaging using plasma polymerization method. The study utilized atmospheric pressure aerosol-assisted plasma deposition to develop a PEG-like coating suitable as a matrix for antimicrobial agents in controlled-release packaging. Results showed optimal plasma operating parameters were essential for desired water stability and biodegradability. Greater distances between the plasma head and substrate led to rougher coatings, while closer distances yielded smoother ones. Low input powers retained more PEG-like chemical structure, although these coatings had less water stability due to low cross-linking. A moderate input power provides the best balance between stability and PEG resemblance.

The reviewer thinks the study is well-organized a well-written, providing good insights and theoretical background. However, as this technology aims to equip the biodegradable property, it would be beneficial to perform any of proof-of-concept experiments to show anything regarding it, at least showing the photos while biodegradation or simple quantitative data.

Author Response

"Response to the comments" (ID: nanomaterials-2607014)

Submission Title: "Degradable plasma-polymerized poly(ethylene glycol)-like coatings as a matrix for food packaging applications"

Dear Editorial Board of the “Nanomaterials,”

We appreciate you for reviewing our manuscript and considering the revised version. We also thank the reviewers for pointing out some important modifications that are essential for our manuscript. To answer their concerns, we have carefully considered all the constructive comments, which have been very useful for restructuring the manuscript. Our point-by-point explanations of the changes made in response to the reviewers’ comments are included in the following pages. We hope that all these changes fulfill the requirements required to deem the manuscript acceptable for publication in nanomaterials.

We look forward to hearing from you in due time regarding our re-submission, and we will respond to any further questions and comments you may have.

Referee: 2

COMMENTS TO THE AUTHOR:

The paper explores an eco-friendly method for producing advanced polymer films for food packaging using plasma polymerization method. The study utilized atmospheric pressure aerosol-assisted plasma deposition to develop a PEG-like coating suitable as a matrix for antimicrobial agents in controlled-release packaging. Results showed optimal plasma operating parameters were essential for desired water stability and biodegradability. Greater distances between the plasma head and substrate led to rougher coatings, while closer distances yielded smoother ones. Low input powers retained more PEG-like chemical structure, although these coatings had less water stability due to low cross-linking. A moderate input power provides the best balance between stability and PEG resemblance.

The reviewer thinks the study is well-organized a well-written, providing good insights and theoretical background. However, as this technology aims to equip the biodegradable property, it would be beneficial to perform any of proof-of-concept experiments to show anything regarding it, at least showing the photos while biodegradation or simple quantitative data.

Reply:

We would like to thank the referee for the positive evaluation.

Thanks for your comment. Indeed, the degradation study is essential for the developed PEG-like coatings. In this work, we studied the stability of the coating over time in an aqua environment (often a case for food storage and packaging). It was shown in Fig.9 that the developed PEG coating is degradable and will be entirely dissolved over time (product shelf life). However, the applied substrate UHMWPE is not biodegradable; therefore, we could not perform the biodegradation test for the whole coating packaging material. 

On the other hand, for the ongoing project, which is the continuation of this work, we are assessing the degradation process of the optimized coating in contact with different food simulants. In addition, we will identify the degradation products. This study is very time and labor-consuming, and we are expecting to get a full set of results in a time span of 6-7 months. If the referee agrees, we will indicate the solubility properties of the coating, emphasize the degradability of the top layer, and emphasize the requirement for a “real life” application to replace UHWMPE, a fully biodegradable substrate (which is also a large game-changing subject in food science nowadays without yet fully defines solution for large scale market).

Reviewer 3 Report

Dear Authors,

Your manuscript is well written and generally scientifically sound. However, the work clearly relates to the plasma treatment of a coating material for food packaging. The scope of the special issue is related to field of plasma technology and nanomaterials and this manuscript has no relevance to the latter. Although you have reported some AFM results (nanoscale roughness values), this coating is not a nanomaterial. The manuscript is clear with few flaws or areas of concern, but I do not believe it fits this particular special issue or any issue of Nanomaterials.

Author Response

"Response to the comments" (ID: nanomaterials-2607014)

Submission Title: "Degradable plasma-polymerized poly(ethylene glycol)-like coatings as a matrix for food packaging applications"

Dear Editorial Board of the “Nanomaterials,”

We appreciate you for reviewing our manuscript and considering the revised version. We also thank the reviewers for pointing out some important modifications that are essential for our manuscript. To answer their concerns, we have carefully considered all the constructive comments, which have been very useful for restructuring the manuscript. Our point-by-point explanations of the changes made in response to the reviewers’ comments are included in the following pages. We hope that all these changes fulfill the requirements required to deem the manuscript acceptable for publication in nanomaterials.

We look forward to hearing from you in due time regarding our re-submission, and we will respond to any further questions and comments you may have.

Referee: 3

COMMENTS TO THE AUTHOR:

Your manuscript is well written and generally scientifically sound. However, the work clearly relates to the plasma treatment of a coating material for food packaging. The scope of the special issue is related to field of plasma technology and nanomaterials and this manuscript has no relevance to the latter. Although you have reported some AFM results (nanoscale roughness values), this coating is not a nanomaterial. The manuscript is clear with few flaws or areas of concern, but I do not believe it fits this particular special issue or any issue of Nanomaterials.

Reply:

We do not fully agree with the referee. In this work, we have demonstrated that the nano-structure of the coatings and chemical composition defined by nanoscale cross-linking of the precursor represents the stability of the deposits. This is an essential link between nanoscience and coatings performance on a macro-scale (stability), which can be necessary for the journal's readers. We also believe that the coating's thickness of 100's nm also belongs to the nanoscale world where, e.g., delamination on the nm scale is a critical nanoscale process, as shown in this work.  

Round 2

Reviewer 1 Report

It can be accepted now

Author Response

We want to thank the referee for the positive evaluation and for considering that our manuscript is acceptable to be published.

Reviewer 3 Report

Dear Authors,

As I previously reported, the manuscript is a good contribution to the area of plasma technology. My main concern was the lack of discussion related to nanomaterials. I agree that there are some aspects related to the nanoscale but you have not implicitly discussed these features in accordance with your author's reply. If you can make some minor discussion points and comments throughout relevant sections, it would become more apparent to the reader that focus includes nanostructures that are applicable to coatings and packaging.

Author Response

We want to thank the referee for the positive evaluation and constructive comments.   

Considering your comment, we added some parts in the manuscript regarding the effect of the nanostructure of the coatings on their physicochemical properties, as you pointed out. 

We explained the nanostructuring influence of the plasma in the introduction, which can be developed during the plasma polymerization and provide beneficial properties due to the extended surface. In addition, the explanation of the effect of created nanoscale roughness on the morphology and the wettability of the developed PEG-like coatings were added in the results and discussion section. Furthermore, the obtained PEG-like coating will be implemented to develop antibacterial packaging for food applications for future analysis. Therefore, we added in the future outlook section that in future work, the antibacterial nanoparticles will be incorporated into the plasma polymerized PEG coatings in order to obtain nanomaterial with antibacterial activity as follows:

Line 89-93:

Plasma polymerization can also be applied to deposit nanostructured coatings which can be of use in different technological applications [19]. Thin nanoscale coatings and nanostructured surfaces produced through plasma polymerization can offer advantages over conventional microscale deposition, such as improved chemical and mechanical properties, water repellency, hardness, corrosion resistance, etc. [20].

Line 323-331:

When the plasma-substrate distance is small, polymerization mainly occurs on the surface of the substrate, which contributes to the formation of a smooth conformal coating with a nanoscale roughness. This obtained nanoscale roughness can be very beneficial for the ultimate application which is to use it as an embedding matrix for antibacterial agents. One of the crucial parameters affecting the antibacterial efficiency is the provided contact surface area with the packed food. In this regard, an increased surface area (resulting from the nanoscale roughness) will enhance the antibacterial activity due to the more pronounced interaction of the bacteria with the surface [41].     

Line 617-618:

The obtained nanoscale roughness enhances the solid-liquid interface area, leading to improved wettability properties and a lower WCA value. 

Line 724-726:

In future work, the degradability of the stable coatings will be further examined and their potential to act as a matrix for antibacterial agents will be explored in detail. 

We hope all these changes fulfill the requirements to deem the manuscript acceptable for publication in nanomaterials.